# Testing Explant Sources, Culture Media, and Light Conditions for the Improvement of Organogenesis in *Pinus ponderosa* (P. Lawson and C. Lawson)

**DOI:** 10.3390/plants12040850

**Published:** 2023-02-14

**Authors:** Alejandra Rojas-Vargas, Ander Castander-Olarieta, Antonia Maiara Marques do Nascimento, María Laura Vélez, Cátia Pereira, João Martins, Mónica Zuzarte, Jorge Canhoto, Itziar A. Montalbán, Paloma Moncaleán

**Affiliations:** 1Instituto de Investigación y Servicios Forestales, Universidad Nacional, Heredia 86-3000, Costa Rica; 2NEIKER-BRTA, Department of Forestry Sciences, 01192 Arkaute, Spain; 3Centro de Investigación y Extensión Forestal Andino Patagónico (CIEFAP), Consejo Nacional de Investigaciones Científicas y Técnicas (CONICET), Esquel 9200, Argentina; 4Departamento de Ingeniería Forestal, Universidad Nacional de la Patagonia (UNPSJB), Ruta 259 Km 16,24, Esquel 9200, Argentina; 5Centre for Functional Ecology, TERRA Associate Laboratory, Department of Life Sciences, University of Coimbra, Calçada Martim de Freitas, 3000-456 Coimbra, Portugal; 6Coimbra Institute for Clinical and Biomedical Research (iCBR), Faculty of Medicine, University of Coimbra, Azinhaga de Santa Comba, 3000-548 Coimbra, Portugal; 7Clinical Academic Centre of Coimbra (CACC), 3000-548 Coimbra, Portugal

**Keywords:** cytokinins, LEDs, micropropagation, rooting, scanning electron microscopy, shoot induction

## Abstract

*Pinus. ponderosa* (P. Lawson and C. Lawson) is a commercial tree and one of the most important forest species in North America. Ponderosa pine suffers hardship when going through vegetative propagation and, in some cases, 15–30 years are needed to achieve full reproductive capacity. Based on previous works on *P. ponderosa* regeneration through in vitro organogenesis and trying to improve the published protocols, our objective was to analyze the influence of different types of explants, basal culture media, cytokinins, auxins, and light treatments on the success of shoot multiplication and rooting phases. Whole zygotic embryos and 44 µΜ 6-benzyladenine showed the best results in terms of explants survival. For shoot organogenesis, whole zygotic embryos and half LP (LP medium, Quoirin and Lepoivre, 1977, modified by Aitken-Christie et al., 1988) macronutrients were selected. A significant positive interaction between whole zygotic embryos and half LP macronutrients was found for the percentage of explants forming shoots. Regarding the light treatments applied, a significantly higher percentage of shoots elongated enough to be rooted was detected in shoots growing under blue LED at a light intensity of 61.09 µmol m^−2^ s^−1^. However, the acclimatization percentage was higher in shoots previously cultivated under fluorescent light at a light intensity of 61.71 µmol m^−2^ s^−1^. Anatomical studies using light microscopy and scanning electron microscopy showed the light treatments promoted differences in anatomical aspects in in vitro shoots; needles of plantlets exposed to red and blue LEDs revealed less stomata compared with needles from plantlets exposed to fluorescent light.

## 1. Introduction

Conifers cover approximately 39% of the world forests and are the best known and most economically important species among gymnosperms; with a total of around 615 species, including pines (*Pinus* spp.) [1,2].

Ponderosa pine (Western Yellow Pine) is the largest of the western pine species and is a major lumber tree in the Western North America; its natural range includes every state west of the Great Plains and north into Western Canada and south into Mexico [3,4]. In the West of the United States, it is considered one the most fire-resistant conifers. This species develops a protective outer corky bark early in life and presents long needles with high moisture content surrounding the terminal buds, which helps to protect the apical meristems, allowing branch tips to refoliate [5,6,7].

The species is a major source of timber; ponderosa pine forests are also important as wildlife habitats, for recreational use, and for esthetic values [3,4]. However, climate change conditions can provoke problems because this species is susceptible to diseases such as Diplodia tip blight (*Diplodia pinea*), Dothistroma blight (*Dothistroma pini*), and Elytroderma needle blight (*Elytroderma deformans*); although some authors consider ponderosa pine to be relatively disease resistant [3,8].

The time taken for conifers to reach sexual maturity is long, often 20–30 years; therefore, considerable time is required before superior offspring appear [3,8]. Grafting, rooting cuttings, and tissue culture are vegetative propagation methods used to improve this process [3]. In this sense, in vitro methods, including somatic embryogenesis or shoot organogenesis followed by rooting, provide valuable tools that can be used for the propagation of ponderosa pine [9]. Quite often, in vitro culture has low success rates of propagation due to the multitude of physico-chemical conditions influencing plant growth. Factors, such as the type and concentration of plant growth regulators, basal media, solidifying agents, type of explant, among others, can be manipulated to induce or optimize the whole process [9,10].

Among these physico-chemical factors, light is one of the most important factors controlling plant growth and development [11]. In general, fluorescent light (FL) has been used as the main source of lighting, with irradiances between 25 and 150 mmol m^−2^ s^−1^ for a 16 h photoperiod [12]. However, the power consumption of this kind of lamp is high, making the process expensive, and culture rooms need to remove the heat emitted by them using air conditioners together with FL efficiency [10]. For these reasons, the use of light emitting diodes (LEDs) is recommended as they present advantages over FL, such as durability, low power consumption, the possibility to fix specific wavelengths, smaller size, and negligible heat production [13].

Previous works dealing with the micropropagation *P. ponderosa* are scarce. Ellis and Biwerback (1984) [8] studied multiple bud formations culturing zygotic embryos. Years later, Ellis and Judd (1987) [14] analyzed protein profiles on bud-forming cotyledons. In 1990, Tuskan et al. [15] investigated the influence of plant growth regulators, basal media and carbohydrate levels. Other authors investigated in vitro formation of axillary buds from immature shoots [16], and the influence of different culture media on bud induction in whole embryos and cotyledons [3]. Finally, the effect of cytokinins on plastid development and photosynthetic polypeptides during organogenesis has also been investigated [17]. 

In our lab, several studies have been developed to optimize somatic embryogenesis [18] and organogenesis [19,20] in many species of *Pinus*. Moreover, as a result of priming cells during the somatic embryogenesis process in conifers, different metabolic, epigenetic, and proteomic profiles related to abiotic stress tolerance have been described recently [21,22,23]. However, as far as we know, no work has been done in order to improve the micropropagation of *P. ponderosa* by testing the chemical composition of the culture media or physical factors, such as type of lights (fluorescent versus LEDs) and their effect on morphogenesis, tissue micromorphology, and the overall success of the process.

Considering the abovementioned information, the main objective of our study was to optimize the organogenesis process of *P. ponderosa* in order to develop an efficient protocol for in vitro propagation. To carry out this objective, we focused on improving (1) the multiplication phase using different types of explants, cytokinins, culture media, and light treatments, and (2) the rooting stage using different auxins and light treatments. 

## 2. Results

### 2.1. Induction of Organogenesis

#### 2.1.1. Experiment 1

Contamination rates were registered after sterilization protocols A (21%) and B (16%). The sterilization protocol did not significantly affect the survival percentage. Based on this result, protocol B was selected as the preferred sterilization method in Experiment 2.

When the type of initial explant was analyzed, a significantly higher survival percentage was observed when using whole zygotic embryos (89%) instead of isolated cotyledons (77%).

Explants’ survival was also significantly affected by the 6-benzyladenine (BA) concentration used (Appendix A); explants cultured in 44 μM BA showed significantly higher survival rates (86%) than those induced at 4.4 μM BA (Figure 1).

The interaction between BA concentration and culture medium, and between explant type and sterilization protocol did not reveal statistically significant differences (Appendix A).

When considering all the variables analyzed (sterilization protocol, explant type, culture medium, and BA concentration) and the interaction between them, after four weeks of induction medium (IM), statistically significant differences were only found for the percentage of explants forming shoots (EFS) depending on the culture medium, the explant type and the interaction between them (Appendix A). Whole zygotic embryos cultured in half LP macronutrients (HLP) showed significantly higher EFS than both the embryos cultured in LP and cotyledons, regardless of the culture medium employed (Figure 2). When we used cotyledon as initial explants, buds developed slowly and failed to elongate. Based on these results, whole zygotic embryos and HLP were selected as the best conditions for shoot organogenesis in Experiment 2. 

The highest number of shoots (3.31 ± 2.8) were found in whole zygotic embryos cultured in HLP supplemented with 22 μM BA. 

#### 2.1.2. Experiment 2

In this experiment, the contamination rate registered was 1%. As shown in Appendix A, statistically significant differences were found for the survival percentage according to cytokinin type and light treatment. However, Tukey’s post hoc test did not detect these differences because the *p*-value was bordering on significance. The survival percentage was 93% in whole zygotic embryos cultured with meta-Topolin (m-T) in the culture medium and growing under blue LEDs. On the other hand, 100% survival was found in whole zygotic embryos growing under the remaining treatments tested.

After five weeks of culture in the induction medium (IM), no significant differences were found for the percentage of EFS according to the cytokinin type, the light treatment, or the interaction between them (Appendix A). The EFS ranged from 43% in whole zygotic embryos induced with m-T under white LEDs to 60% in whole zygotic embryos cultured with BA under blue and red LEDs, and whole zygotic embryos induced with m-T under FL.

Regarding the number of shoots per explant (NS/E), no significant differences were observed for the variables studied (cytokinin type and light treatment), or the interaction between them (Appendix A). The NS/E produced ranged from 5.40 ± 1.07 in whole zygotic embryos cultured with 13.1 μM m-T under FL to 9.11 ± 1.55 in whole zygotic embryos induced with 13.1 μM BA and exposed to red LEDs. However, at the end of the elongation stage of shoot organogenesis, a significantly higher percentage of shoots elongated enough to be rooted (PSR) was obtained in explants developed under blue LEDs (48%) than in those exposed to red LEDs (16%); explants grown under white LEDs or FL showed intermediate values (Figure 3).

On the other hand, the cytokinin type and the interaction between cytokinin type and light treatment did not have a significant effect on the PSR (Appendix A). 

No statistically significant differences were observed in shoots exposed to different light treatments for root induction percentage (RI), number of roots per explant (NR/E), or the length of the longest root (LLR) (Appendix A). The RI percentage ranged from 3% in shoots cultured with 5 µM 1-naphthaleneacetic acid (NAA) to 18% in shoots grown with a mixture of 5 µM NAA and 5 µM indole-3-butyric acid (IBA). Shoots induced with 10 µM NAA showed an intermediate value of RI (12%). 

The highest response for root induction was obtained in shoots exposed to white LEDs and the lowest response was recorded in shoots under blue LEDs (Table 1). When shoots were exposed to white LEDs, the highest NR/E was obtained, and the lowest response was found in shoots under red LEDs. The longest primary roots (ranging from 2.78 to 2.15) were recorded in shoots exposed to white LEDs and FL, respectively. Shoots growing under red LEDs displayed the lowest response (Table 1).

Acclimatized shoots were successfully obtained from shoots exposed to FL (50%), shoots under white LED (20%), and shoots exposed to blue LEDs (14%); it was not possible to acclimatize shoots grown under red LED.

### 2.2. Morphological Characterization of Needles

The effect of the chemical composition of the culture media and the physical factors on tissue micromorphology from needles of in vitro plantlets was studied, and then it was compared to the micromorphology from needles of ex vitro plants. In this way, adaxial and cross sections of *P. ponderosa* needles from plantlets cultured with different cytokinin types and light treatment were analyzed. Regardless of the cytokinin type employed, needles from plantlets did not show microscopic alterations in the internal structure and organization.

Adaxial needle surfaces from in vitro plantlets indicated the presence of stomata arranged in parallel and uniseriate bands, while the margin areas were serrated with spinose teeth (Figure 4a). 

The longitudinal section of needles showed the parenchyma mesophyll cells and the large intercellular spaces between them. No structural changes in the mesophyll cell of needles from plantlets exposed to light treatments were found (Figure 4b). Furthermore, in the center of the needle cross section, differentiated vascular bundles with phloem and xylem were identified (Figure 4c).

Histological analysis of needles from in vitro plantlets showed that the epidermis presented a single-cell layer and numerous tannin-rich cells were spread among the mesophyll, where large intercellular spaces were common (Figure 4d). No clear differentiation of the endodermis and the vascular bundle was observed (Figure 4d).

As a general trend, needles of plantlets exposed to blue and red LEDs showed less stomata compared with the stomata of needles from plantlets grown under FL (Figure 5a,c,h). In terms of arrangement and number of stomata, when the adaxial surface of a needle growing under FL was analyzed, a regular pattern in the needle was found (Figure 5h), such as mentioned above (Figure 4a).

The morphology of the stomata of needles from shoots grown under white LEDs (Figure 5e) was similar in the pattern of arrangement to those coming from needles of shoots grown under FL (Figure 5h). Regarding their shape, stomata were ovoidal in all light conditions. The scanning electron micrograph of adaxial needle surfaces showed spinose teeth at the margins (Figure 5a,c,e,f,h). No microscopic alterations in the morphological characterization of the spinose teeth of needles from plantlets grown under different light treatments were observed (Figure 5a,c,e,h).

The internal structure and organization of needles from plantlets was similar for all light treatments (Figure 5b,d,g,i).

SEM analysis showed similar anatomical aspects when needles from field-growing plants were compared with needles of in vitro plantlets (Figure 6a). The characterization of needles from field-growing plants revealed that the epidermal cells were arranged in a tightly packed single layer and the presence of a thin layer of cuticle on their outer surfaces was also observed (Figure 6a).

The middle zone showed the organization of the vascular bundle and surrounding tissues; xylem, phloem, endodermis, and transfusion tissue were recognizable (Figure 6a).

The resin duct was generally localized adjacent to the epidermis and partially immersed in the mesophyll cells (Figure 6b). The presence of starch grains in the parenchyma cells (transfusion tissue) surrounding the vascular bundles was also distinguished (Figure 6c). The cell’s morphology revealed the presence of ridge-like invaginations typical of the cell walls of the mesophyll parenchyma cells and the hypodermis was also identified (Figure 6d).

## 3. Discussion

The establishment of an aseptic in vitro culture is one of the most important factors of plant tissue culture and the effective elimination of contamination contributes to the better development of the explants [26,27]. When introducing *P. ponderosa* material in vitro, contamination rates ranging from 1% to 16% were obtained using hydrogen peroxide (protocol B). In *P. radiata* [20,28], hydrogen peroxide was an effective agent for surface-sterilization of seeds, obtaining contamination percentages below 7%. Our results suggest that the protocol used in this work was adequate for this species and can be used routinely, being a valid procedure for the propagation of other *Pinus* species as well. The most important initial step in the micropropagation process is an adequate sterilization protocol that minimizes tissue damage to achieve the greatest number of aseptic explants ready to be used in the next phases of micropropagation [27,29].

As has been recently reviewed, factors such as explant source and plant growth regulators influence the in vitro regeneration of plants via organogenesis in different species [30,31,32]. In this regard, other authors mentioned that cotyledon explants had several advantages over embryonic explants [33,34]. First, when using cotyledons, the explant area that is exposed to the culture medium is bigger than when culturing whole zygotic embryos. Second, most seeds have at least eight cotyledons, which could be distributed among the various treatments [33,34]. On the contrary, in *P. halepensis* Mill. when using whole embryos or cotyledons as initial explants, over 90% of the embryos gave rise to adventitious buds, whereas isolated cotyledons slowly developed adventitious buds and these buds failed to elongate [35]. These results are in accordance with our study. When we used cotyledon as initial explants, buds developed slowly and failed to elongate. Additionally, De Diego et al. [36] selected embryos as initial explants for *P. pinaster* and mentioned that this would increase the genetic diversity as well as considerably decrease the manual labor cost.

In our study, the capacity of explants to form shoots was significantly higher when whole zygotic embryos were cultured in HLP. Following the trend observed in previous studies [3,8,14], significant differences in shoot induction rates were found when different culture media were tested for induction in *P. ponderosa*. Furthermore, several studies in *P. halepensis*, *P. pinaster*, *P. ayacahuite*, and *P. ponderosa* reported that salt concentration in the culture medium had an influence on morphogenesis [14,34,35,36].

In woody species, it has been recognized that cytokinins are necessary to promote in vitro multiple shoot formation [36,37]. Among cytokinins, BA is the most commonly used in plant tissue culture due to its effectiveness and affordability [38].

In our experiments, BA at the concentrations tested (4.4, 22, and 44 µM) showed efficient organogenic response for NS, and, although not significant, a slightly higher response in whole zygotic embryos cultured in HLP supplemented with 22 µM BA was observed. A similar tendency was observed in several micropropagation protocols in *Pinus* where the culture medium was supplemented with BA (from 1 to 50 µM) to obtain bud induction [19,27,39,40,41,42].

When EFS and NSE were analyzed, no significant differences were found for the effect of BA or m-T, but a slightly higher response in explants cultured with BA was observed. BA in in vitro culture can cause hyperhydricity, shoot-tip necrosis, and histogenic instability [43]. This was not the case in our work, where BA did not provoke in vitro abnormalities. In this sense, in the last years, many reports have described the use of topolins in plant tissue culture, because topolins improve parameters, such as multiplication rate, alleviating physiological disorders and increasing the rooting percentage [44]. In contrast to our study, in *Aloe polyphylla*, the use of m-T resulted in a high multiplication rate [38]. Likewise, Werbrouck et al. [45] found better shoot–root balance when m-T was used in *Spathiphyllum floribundum*. In the same way, in *P. sylvestris* and *P. pinaster*, a higher rate of organogenic response was obtained by using m-T instead of BA [36,46]. However, in other studies of our group, no improvement was found when using m-T instead of BA for the induction of axillary shoots in *P. radiata* and *Sequoia sempervirens* [20,32,41]. Summarizing, the effect of different cytokinins at different concentrations depends on the species and the initial explant used to initiate the in vitro culture.

The influence of light in regulating growth, development, and adventitious root formation has been confirmed in different studies [43,47]. LED showed positive effects in in vitro plants responses, when used as an alternative to conventional lighting [43,48]. Several studies have indicated the stimulatory effect of red or blue LEDs on shoot organogenesis [49]. In this work, no statistically significant difference was found in EFS when whole zygotic embryos were cultured under different light treatments. In contrast to our work, in *Gerbera jamesonii* in vitro shoots growing under red LEDs displayed the greatest elongation rates [50]. Chen et al. [43] in *Passiflora edulis* reported an increase in chlorophyll content and plantlet quality when plants were grown under red LEDs. Additionally, the highest stem length was observed in in vitro potato plantlets under red LEDs [51]. 

When studying the effect of the light treatment on PSR, shoots growing under blue LEDs showed the highest response. These results agree with those observed in *Curculigo orchioides* and *Zingiber officinale* where blue LEDs promoted in vitro shoot formation [52,53]. Furthermore, Chen et al. [51] observed the greatest stem diameter and the highest health index in *Solanum tuberosum* when cultured under blue LEDs. 

Studies performed in *Z. officinale* and *Phalaenopsis pulcherrima* indicated that LED treatments promoted in vitro root induction and development [53,54]. In our experiments, when the LED was tested, no significant differences were found for the RI, NR/E, and LLR, but a slightly higher response in shoots under white LEDs was found. In contrast, in *Z. officinale* and *Citronella mucronata* a higher rooting response was found when cultured under red or the combination of red and blue LEDs in a ratio of 1:2 [53,55].

In *Cucumis metuliferus,* explants showed high root induction rates when exposed to white LEDs and in a mixed circuit of blue and red [56]. Furthermore, *Cedrela odorata* in vitro cultures grown under white LEDs presented a higher number of roots formed [47]. In this regard, Lai et al. [56] explained that one reason for this result might be that white light provides with a wider variety of minor lights, which are also required by plants for optimal growth. 

Contrary to our results, in *P. radiata*, in vitro shoots growing under FL showed high NR/E [42]. Regarding LLR, shoots under white LEDs and FL light showed longer primary roots. A similar pattern was reported in *Handroanthus ochraceus* where plants showed higher root length values when exposed to low FL irradiances [12]. In contrast, longer roots were obtained in plants of *P. radiata* and *Vitis ficifolia* growing under red LEDs [42,57].

The light treatments tested promoted differences in anatomical aspects of the stomata in the needles of *P. ponderosa* in vitro shoots. The morpho-anatomic characteristics of needles from in vitro and ex vitro plants were similar and are in accordance with the study described for *P. halepensis* by [58].

The teeth structures on the needle surface were morphologically similar to those found in primary needles of *P. halepensis* and *P. nigra* [58,59]. In this sense, Boddi et al. [58] described teeth or spine-like structures in transverse sections as more elongated cells with cytoplasmic content differing from the adjacent epidermic cells for thicker walls; however, the relevance of these structures is unknown. A scanning electron micrograph revealed stomata arranged in parallel and as uniseriate bands, which is in accordance with the work described for *P. nigra* by Mitic et al. [59]. Similar to that observed in *P. halepensis*, [58] large and ovoidal stomata were detected in this study. In contrast, stomata in primary needles of *P. canariensis* presented a volcanic shape [60].

SEM analysis revealed the presence of hypodermis in needles from field-growing plants. Contrary to our results, in *P. halepensis* seedlings, primary needles were characterized by the absence of a hypodermal cell layer and the epidermis being in direct contact with the mesophyll tissue, but these differences could be attributed to the fact that these analyses were performed in seedlings 22–24 weeks after emergence [58]. 

LED sources improve the leaf anatomy of in vitro plantlets in *Musa* spp.; the abaxial epidermis was thinner than the adaxial epidermis [61]. In *Chrysanthemun* leaves, the thickness of the adaxial epidermal cells increased under white LED; blue LED favored the anatomical development of the palisade parenchyma layer; and red LED reduced this parenchyma [62]. Red LED also affected the epidermis morphology in *Solanum tuberosum* plantlets. Epidermises with greater thickness were induced and palisade parenchyma and spongy parenchyma were arranged neatly [51]. However, LED illumination did not improve morpho-anatomic characteristics, including vascular bundles from in vitro cultures of *P. ponderosa*. 

Silva et al. [63], in *Pfaffia glomerata* in vitro plantlets, increased in the size of vascular bundles and vessel elements under combinations of red and blue LED. In the same way, the central vascular bundle of plants from *Epidendrum fulgens* grown with natural ventilation under combinations of white/high blue, deep red/white/medium blue, and deep red/white/far red/medium blue LED had a higher content of sclerenchyma [64]. Furthermore, Gnasekaran et al. [65] found that LED spectral quality alters plant chloroplast ultrastructure through the effects on starch accumulation in *Z. officinale*. Summarizing, the effect of different LEDs sources at different concentrations in morpho-anatomic characteristics seems to depend on the species cultured at in vitro conditions.

All these aspects could, to some extent, explain the results obtained in our study, where the basal media, cytokinins, the explant type, and the light quality influenced the organogenic process and root formation of *P. ponderosa* and their ex vitro acclimatization. Furthermore, our results demonstrated that different LEDs can be strategically used to improve micropropagation efficiency and reduce the costs of in vitro ponderosa pine plant production.

## 4. Materials and Methods

### 4.1. Plant Material

*P. ponderosa* (P. Lawson and C. Lawson) seeds were obtained from 14-year-old trees grown in a clonal seed orchard of the Rotonda, Trevelin Experimental Field, (Argentina; 43°06′05″ N, 71°33′30″ W) (batch number: 25 U7519 JP) and they were also purchased from the Instituto Nacional de Tecnología Agropecuaria (Trevelin, Argentina). 

### 4.2. Sterilization

Seeds were rinsed under running water for 5 min and sterilized following two different protocols: (1) commercial bleach 5% (active chlorine 37 gL^−1^ sodium hypochlorite) plus one drop of Tween 20^®^ for 20 min, followed by three rinses in sterile distilled water for 5 min each; (2) 10% H_2_O_2_ (*v/v*) (30% active H_2_O_2_) plus one drop of Tween 20^®^ for 20 min, followed by three rinses in sterile distilled water for 5 min each. The sterilization protocols were performed under sterile conditions in a laminar flow unit. Seeds were stored in moistened sterile filter paper at 4 °C in darkness. After four days, seed coats and megagametophytes were removed aseptically and two explant types (a) cotyledons (Figure 7a) or whole zygotic embryos (Figure 7b) were used.

### 4.3. Organogenic Process

#### 4.3.1. Experiment 1

After sterilization, explants (cotyledons or whole zygotic embryos) were cultivated on Petri dishes (90 × 15 mm) containing 20 mL of bud induction medium (IM) (Figure 7a,b). Two basal media were assayed: LP (Quoirin and Lepoivre [24] modified by Aitken-Christie et al. [25]) and HLP, which consisted of LP with half macronutrients. Both media were supplemented with 3% (*w/v*) sucrose and solidified with 8 gL^−1^ Difco Agar granulated. Moreover, three concentrations of BA (4.4, 22, and 44 μM) were evaluated. The pH of all media was adjusted to 5.8 before autoclaving (121 °C, 20 min). Cotyledons were excised from embryos and placed horizontally onto the induction medium; whole zygotic embryos were cultured in an inverted position with the cotyledons immersed in the induction medium. 

As soon as bud induction was observed (after four weeks), the explants were transferred to Petri dishes containing 20 mL of elongation medium (EM). EM consisted of hormone-free LP or HLP supplemented with 2 gL^−1^ activated charcoal, 3% (*w/v*) sucrose, and was solidified with 8.5 gL^−1^ Difco Agar granulated. After 30 days in culture (when elongating needle fascicles were evident), explants were subcultured into baby food jars with Magenta TM b-cap lids containing 25 mL of LP or HLP elongation medium. The shoots were subcultured every six weeks into the same medium. When shoots were 10–15 mm long, they were separated and cultivated individually in a fresh medium. 

All the cultures were laid on the growth chamber at a temperature of 21 ± 1 °C, at a 16 h photoperiod with 120 µmol m^−2^ s^−1^ light intensity provided by cool white fluorescent tubes (TLD 58 W/33; Philips, France).

#### 4.3.2. Experiment 2

Based on the results of Experiment 1, sterilization protocol B and whole zygotic embryos were used as initial explant and HLP was selected to carry out this experiment. Two types of cytokinins were tested: BA and m-T at the same concentration (13.1 µΜ). Three different light treatments were also tested for the conditions named above: (A) blue light (peak wavelength 470 nm), 61.09 µmol m^−2^ s^−1^; (B) red Light (peak wavelength 630 nm), 61.09 µmol m^−2^ s^−1^; and (C) white light (color temperature 4000 K), 61.09 µmol m^−2^ s^−1^, all of them provided by adjustable LEDs (RB4K Grow LEDs). As a control, whole zygotic embryos were cultured under cool white fluorescent light (FL) (TLD 58 W/33; color temperature 4100 K) at 61.71 µmol m^−2^ s^−1^ light intensity. The cultures were laid on the growth chamber at a temperature of 21 ± 1 °C for five weeks at a 16 h photoperiod.

As soon as bud induction was observed (after four weeks) (Figure 7c), the explants were transferred to Petri dishes with HLP elongation medium. After 30 days in culture (when elongating needle fascicles were evident), explants were subcultured into baby food jars with Magenta TM b-cap lids containing 25 mL of the same medium (Figure 7d). The shoots were subcultured every six weeks into the same medium. When shoots were 10–15 mm long, they were separated and cultivated individually. The different lighting conditions described previously were maintained in this phase.

#### 4.3.3. Rooting and Adaptation

After the elongation phase, shoots of at least 20–25 mm long were used for root induction. The explants were cultivated into baby food jars with Magenta TM b-cap lids containing 25 mL of root induction medium, which consisted of HLP supplemented with three different auxins: (A) 5 µMNAA; (B) 10 µM NAA; and (C) a mixture of 5 µM NAA and 5 µM IBA; all media were supplemented with 3% (*w/v*) sucrose and solidified with 8.5 gL^−1^ Plant Agar^®^. The different light conditions were the same as described above. After four weeks of culture in medium A or two weeks in medium B and C, explants were cultured in baby food jars with Magenta TM b-cap lids containing 25 mL of root expression medium (REM), which consisted of hormone-free HLP supplemented with 2 gL^−1^ activated charcoal, 3% (*w/v*) sucrose, and 8.5 gL^−1^ Plant Agar^®^ for six weeks. Growth chamber temperature and photoperiod were the same as those described above.

After six weeks in REM, explants with visible roots (Figure 7e) were transferred to wet peat moss (Pindstrup, Ryomgård, Denmark): vermiculite (8:2, *v/v*) and acclimatized in the greenhouse under controlled conditions at 21 ± 1 °C with progressively decreasing humidity for one month from 95% to 80% (Figure 7f). Prior to acclimatization, and following the procedure described in Castander et al. [66], the plants that developed a poor root system were transferred to Ecoboxes (Eco2box/green filter: a polypropylene vessel with a “breathing” hermetic cover, Duchefa^®^, Duchefa Biochemie, Haarlem, Netherlands) containing perlite:peat (1:1, *v/v*) moistened with liquid HLP. 

### 4.4. Morphological Characterization of Needles

#### 4.4.1. Light Microscopy Analysis

For structural analysis after eighteen weeks in the elongation medium, 16 needles were used following the procedure described in Giacomolli et al. [67]. Two needles per cytokinin and light treatment were fixed in 0.2 M phosphate buffer (pH 7.2) and 2.5% paraformaldehyde for 48 h at 4 °C. Then, the samples were washed twice with 0.1 M phosphate buffer (pH 7.2) for 15 min and then dehydrated in ethanolic series (30%, 50%, 70%, 80%, 90%, 95%, and 100% *v/v*) (1 h each).

Subsequently, the samples were embedded in paraffin by means of dehydration with ethanol and Clear Rite™ and the consecutive immersion in liquid paraffin at 65 °C, as described in Rossi et al. [68]. The dehydration process was the following: ethanol 70% (*v/v*) for 120 min; ethanol 90% (*v/v*) for 90 min (2×); ethanol 95% (*v/v*) for 90 min; ethanol 100% (*v/v*) for 90 min (×2); ethanol: Clear Rite™ (1:1 *v/v*) and pure Clear Rite™ for 90 min (2×). Then, samples were embedded in paraffin wax at 65 °C for 120 min (2×) before the inclusion in paraffin molds. After paraffin inclusion (HistoDream EW, Milestone Medical Sorisole, Italy), sections of 8–10 µm were obtained in a rotary microtome (Microm HM 340E, Thermo Scientific, Waltham, USA) and transferred to microscope slides previously prepared with albumin glycerol and kept at 30 °C for 12 h. Next, the deparaffinization was performed with Clear Rite™ for 20 min, 100% ethanol for 4 min, and washing under running water. The samples were stained with an Astra Blue (0.15%) and Safranin (0.04%) water solution for 10 min; and finally, they were observed in an optical microscope (Leica DM 4000 B, Mannheim, Germany) and photos were taken with a Leica camera (Leica Application Suite version 4.13).

#### 4.4.2. Scanning Electron Microscopy

For scanning electron microscopy (SEM), the analyses were performed following the procedure described in Giacomolli et al. [67]. Two needles per cytokinin and light treatment and two from adult trees were fixed in 0.1 M phosphate buffer (pH 7.2) and 2.5% paraformaldehyde for 24 h at 4 °C. After that, the samples were washed twice with 0.1 M phosphate buffer (pH 7.2) for 15 min and then dehydrated in a series of ethanol at different concentrations (30%, 50%, 70%, 80%, 90%, 95%, and 100% *v/v*) for 15 min each. 

All the samples were cut into small pieces and prepared following the procedure described in Marques et al. [69]. Then, the samples were placed on carbon stickers above metallic stubs, observed without further preparations in freeze conditions (−20 °C) at 10.0 kV, and were analyzed and photographed using a variable pressure scanning electron microscope (Flex SEM 1000, Hitachi, Tokyo, Japan).

### 4.5. Data Collection and Statistical Analysis

#### 4.5.1. Experiment 1

Three Petri dishes and eight to ten explants per Petri dish (cotyledons or whole zygotic embryos) per sterilization protocol, were cultured in each culture medium and BA concentration. Contamination, survival, and explants forming shoots (EFS) percentages for each condition tested were measured after two months of culture. When the axillary shoots were isolated and cultured individually in an elongation medium, the EFS and the mean number of shoots were calculated with respect to the non-contaminated explants. A logistic regression model was used to analyze the effect of the sterilization protocol, explant type, culture medium, and BA concentration on survival and EFS (%). When necessary, Tukey’s post hoc test (α = 0.05) was used for multiple comparisons.

#### 4.5.2. Experiment 2

Five Petri dishes and eight whole zygotic embryos per Petri dish, cytokinin type, and light treatment were cultured in HLP. When the axillary shoots were isolated and cultured individually in the elongation medium, the EFS, the NS/E, and percentage of (PSR) out of the total number of shoots produced per explant were calculated. A logistic regression model was used to analyze the effect of the cytokinin type and light treatment on survival and EFS. When necessary, Tukey’s post hoc test (α = 0.05) was used for multiple comparisons.

The confirmation of the homogeneity of variances and normality of the data on the NS/E and PSR were performed, and PSR was $\sqrt{x}$ transformed to meet homocedasticity. Data for the NS/E and PSR were analyzed by analysis of variance (ANOVA). When necessary, multiple comparisons were made using Tukey’s post hoc test (α = 0.05). 

A completely randomized design was carried out using six to twenty plantlets per cytokinin type and light treatment per each auxin treatment. The RI percentage (RI), the mean NR/E, and the LLR (cm) were recorded after six weeks of culture in a root elongation medium.

To assess the effect of the cytokinin and light treatment on the RI, a logistic regression was performed. Data for NR/E and LLR were analyzed by ANOVA. When necessary, multiple comparisons were made using Tukey’s post hoc test (α = 0.05). The acclimatization percentage was calculated after four weeks under ex vitro conditions. The data were analyzed using R Core Team software^®^ (version 4.2.1, Vienna, Austria).

## 5. Conclusions

The regeneration of *P. ponderosa* through organogenesis using whole zygotic embryos as explants was achieved. The best shoot induction percentage was obtained when whole zygotic embryos were cultured in half LP macronutrients. Following the novel use of LED lights for organogenesis, a higher number of rootable shoots were obtained when blue light was used. However, our results suggest that white LED could be more beneficial for rooting stages. Finally, in the greenhouse, the shoots cultured under fluorescent light showed the highest acclimatization success (50%). This may indicate that the use of different kinds of lights could be necessary during the process in order to optimize the organogenic process in *P. ponderosa*. Concerning the anatomy studies, this was the first microscopy assessment of in vitro plantlets exposed to LEDs treatments. The needles from in vitro and ex vitro plants showed similar morpho-anatomic characteristics. 

It would be interesting for future experiments to test other combinations and concentrations of auxins for rooting induction, and other LEDs, such as a combination of red and blue LEDs, to improve the rooting and acclimatization of shoots. 

## Figures and Tables

**Figure 1 plants-12-00850-f001:**
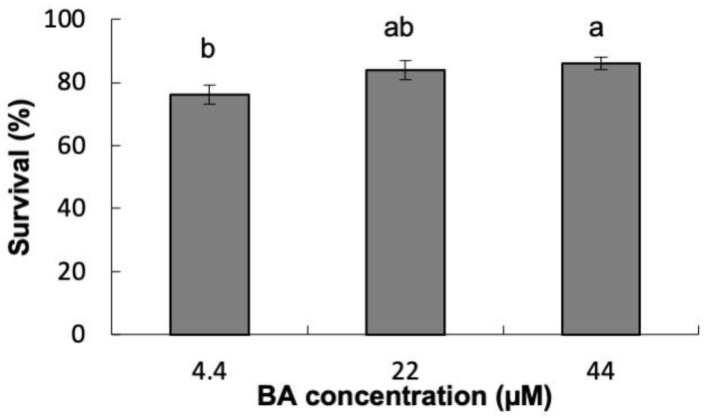
Percentage of survival of *Pinus ponderosa* explants cultured in LP or half LP macronutrients (Quoirin and Lepoivre [24], modified by Aitken-Christie et al. [25]), supplemented with 6-benzyladenine. Data are presented as mean values ± S.E. Different letters indicate significant differences according to Tukey’s post hoc test (*p* < 0.05).

**Figure 2 plants-12-00850-f002:**
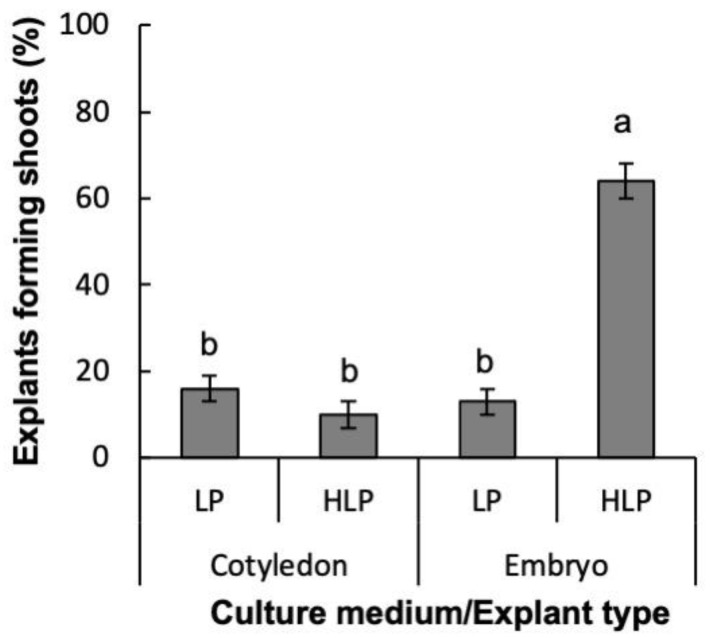
Percentage of explants forming shoots of *Pinus ponderosa* cotyledons or whole zygotic embryos cultured in LP or half LP macronutrients (HLP). Data are presented as mean values ± S.E. Different letters indicate significant differences according to Tukey’s post hoc test (*p* < 0.05).

**Figure 3 plants-12-00850-f003:**
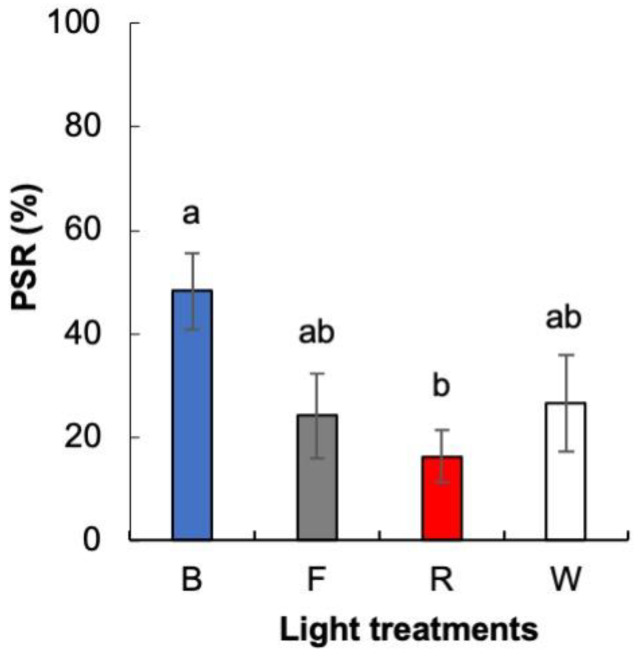
Percentage of shoots elongated enough to be rooted (PSR) of *Pinus ponderosa* growing on half LP macronutrients under light treatments (blue LEDs (B), fluorescent light (F), red LEDs (R), and white LEDs (W)). Data are presented as mean values ± S.E. Different letters indicate significant differences according to Tukey’s post hoc test (*p* < 0.05).

**Figure 4 plants-12-00850-f004:**
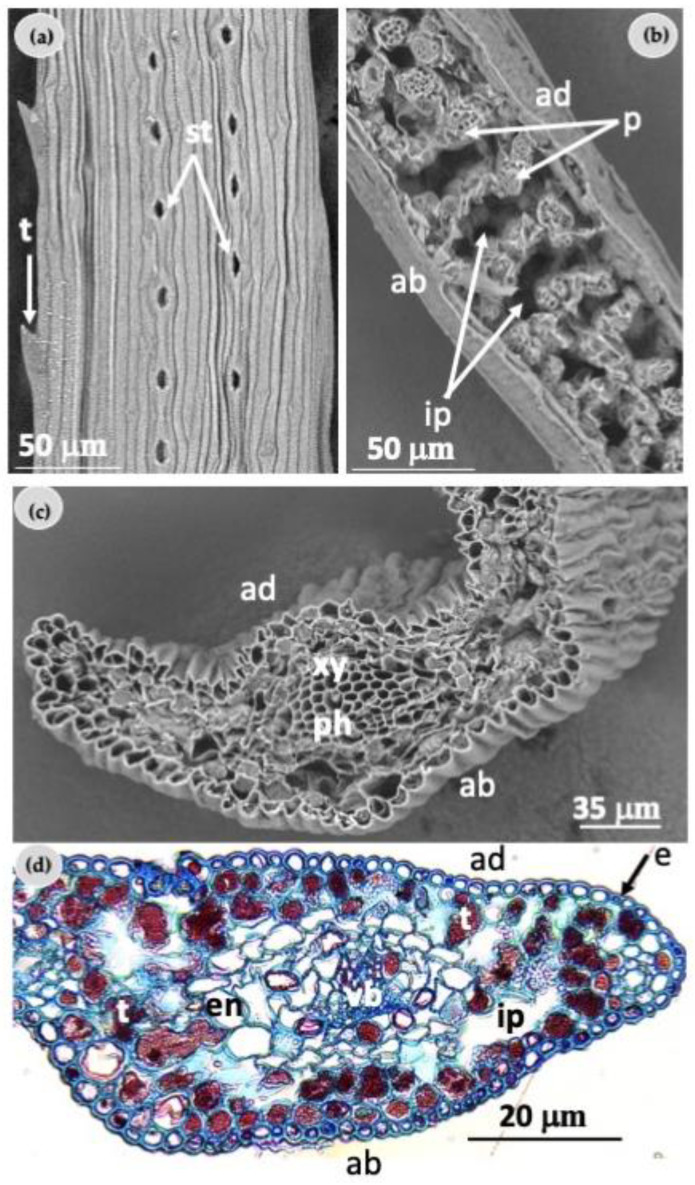
Scanning electron microscopy and histological observations of needles from in vitro *Pinus ponderosa* plantlets: (**a**) adaxial surface of a needle showing teeth (t) and two rows of stomata (st); (**b**) longitudinal section of a needle showing the parenchyma mesophyll cells (p) and the large intercellular spaces (ip) between them; (**c**) SEM observation of a needle cross section; (**d**) histological observation of needle cross section. There is poor differentiation of the vascular bundle (vb), phloem (ph), and xylem (xy) as well as of the endodermis (en). Tannin-rich cells (t) are spread among the mesophyll and large intercellular spaces (ip) are common. ab—abaxial surface; ad—adaxial surface; e—epidermis.

**Figure 5 plants-12-00850-f005:**
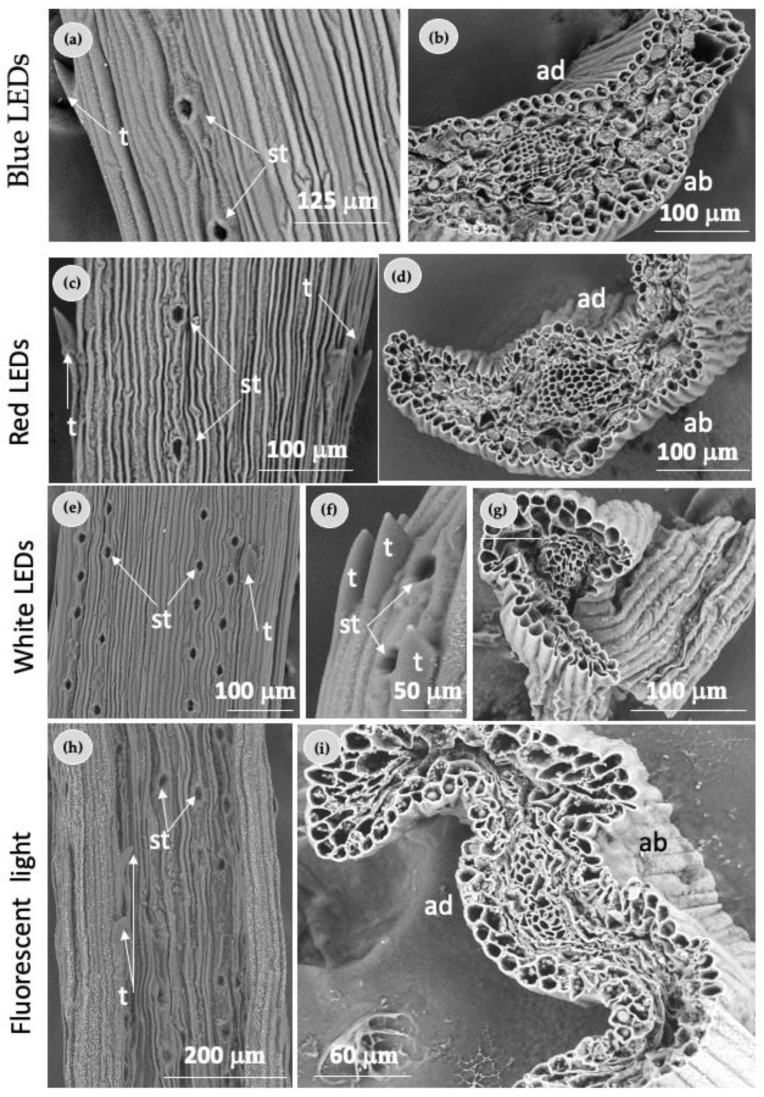
Scanning electron microscopy observations of needles from in vitro *Pinus ponderosa* plantlets exposed to different light treatments: (**a**) adaxial surface of a needle showing teeth (t) and two rows of stomata (st); (**b**) needle cross section; (**c**) adaxial surface of a needle showing teeth (t) and two rows of stomata (st); (**d**) needle cross section; (**e**) adaxial surface of a needle showing teeth (t) and two rows of stomata; (**f**) close view of the stomata. A tooth can also be observed; (**g**) needle cross section; (**h**) adaxial surface of a needle showing teeth (t) and two rows of stomata (st); (**i**) needle cross section. ab—abaxial surface; ad—adaxial surface.

**Figure 6 plants-12-00850-f006:**
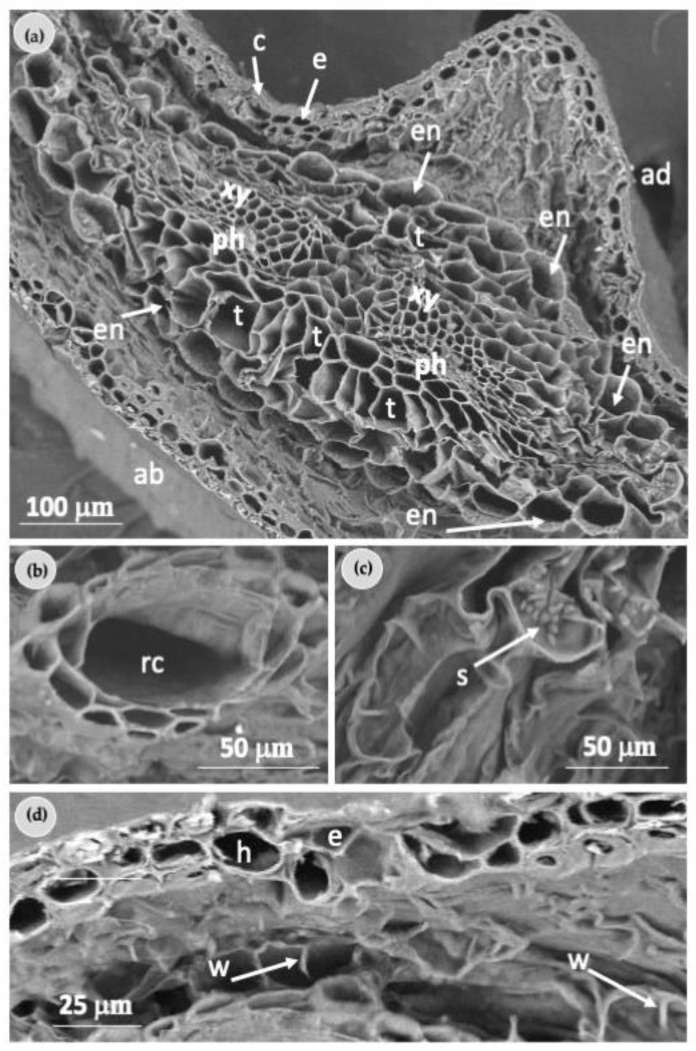
Scanning electron microscopy details of the organization of *Pinus ponderosa* needle from field-growing plants: (**a**) middle zone showing the organization of the vascular and surrounding tissues; (**b**) higher magnification of a resin duct (rc); (**c**) starch grains (s) in the parenchyma cells (transfusion tissue) surrounding the vascular bundles; (**d**) ridge-like invaginations (w) typical of the cell walls of the mesophyll parenchyma cells. ab—abaxial surface; ad—adaxial surface; c—cuticle; e—epidermis; en—endodermis; h—hypodermis; ph—phloem; rc—resin duct; t—transfusion tissue; xy—xylem.

**Figure 7 plants-12-00850-f007:**
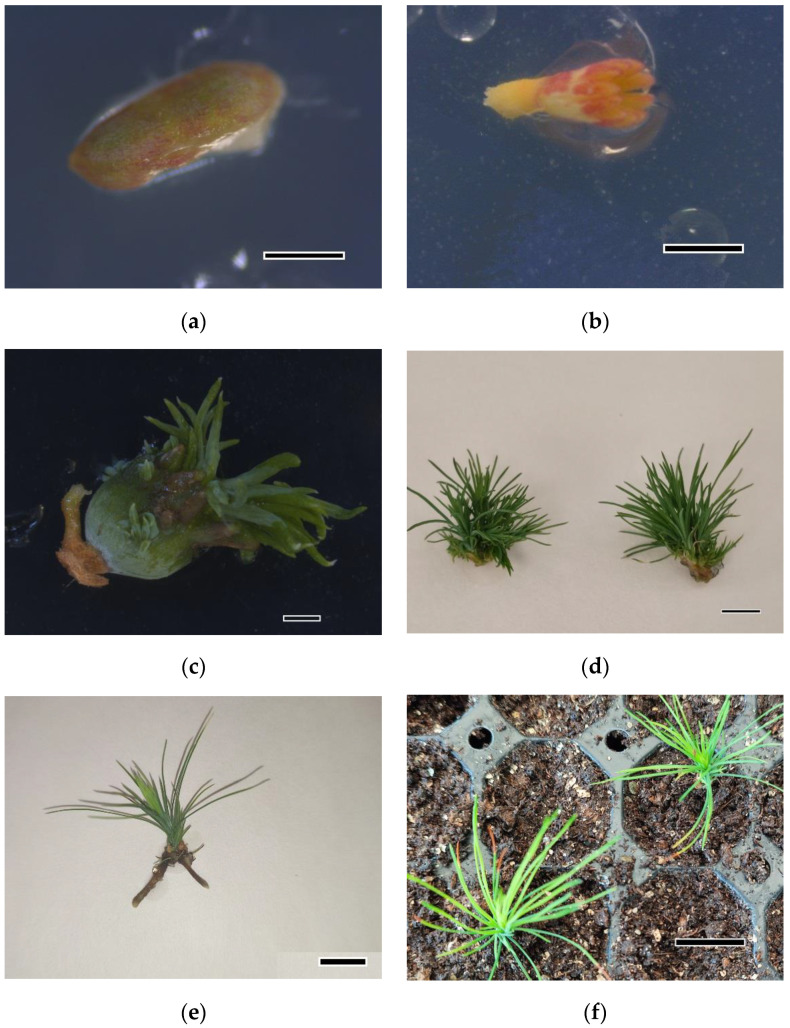
Plant material at different stages of *Pinus ponderosa* organogenic process: (**a**) cotyledon cultured for 2 weeks in LP supplemented with 22 μM 6-benzyladenine (BA), bar = 1.5 mm; (**b**) whole zygotic embryo cultured for 1 week in LP supplemented with 22 μM BA, bar = 1.5 mm; (**c**) developing shoots from zygotic embryo cultured for 4 weeks in half LP macronutrients supplemented with 13.1 μM BA and exposed to red LEDs, bar = 3.0 mm; (**d**) shoots after 6 weeks in elongation medium, which consisted of hormone-free LP supplemented with 2 gL^−1^ activated charcoal, bar = 5.0 mm; (**e**) rooted shoots after 2 weeks in root induction medium, which consisted of half LP macronutrients supplemented with a mixture of 5 µM 1-naphthaleneacetic acid (NAA) and 5 µM indole-3-butyric acid (IBA), bar = 5.0 mm; (**f**) acclimatized shoots exposed to fluorescent light in in vitro conditions after 4 weeks in ex vitro conditions in the greenhouse, bar = 20 mm.

**Table 1 plants-12-00850-t001:** Root induction (%), number of roots per explant, and length of the longest root of *Pinus ponderosa* shoots cultured in half LP macronutrients according to light treatment (blue LEDs, red LEDs, white LEDs, and fluorescent light). Data are presented as mean values ± S.E.

Light Treatment	Root Induction (%)	Number of Roots per Explant	Length of the Longest Root (cm)
Blue LEDs	7 ± 30	1.71 ± 0.29	1.60 ± 0.77
Red LEDs	8 ± 40	1.25 ± 0.25	0.98 ± 0.34
White LEDs	19 ± 80	2.40 ± 0.87	2.78 ± 1.64
Fluorescent	8 ± 40	2.00 ± 0.71	2.15 ± 1.35

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
