# Peer review of "Testing Explant Sources, Culture Media, and Light Conditions for the Improvement of Organogenesis in Pinus ponderosa (P. Lawson and C. Lawson)"

_plants, 2023, doi:10.3390/plants12040850_

Round 1
Reviewer 1 Report
Ponderosa Pine (Western Yellow Pine) is the largest of the western pine species and is major lumber tree in the western North America. The species is a major source of timber, ponderosa pine forests are also important as wildlife habitat, for recreational use, and for esthetic values. The time taken for conifers to reach sexual maturity is known to be long, often 20-30 years; therefore, considerable time is required before superior offspring appear. Grafting, rooting cuttings and tissue culture are vegetative propagation methods used to improve this process. In this regard, the research done in the presented manuscript merits attention.
The article follows the order required by the journal: Introduction, Results, Discussion, Materials and Methods, Conclusions. Abstract is informative; the Introduction is good and to the point to the purpose; The goal is precisely and clearly formulated. The bibliography is actual. Over 50% of the cited literature is from the last 10 years, оf which over 90% are from the last 5 years. However, the quality of the manuscript will improve if the authors use clearer and more precise sentences, as they are difficult to understand. It is essential to improve “Material and Methods” as well as the data presentation. In my opinion, MM should be reworked, with some of the points noted below.
The authors should carefully and critically review publications on the subject and highlight clear their novel contribution.
I have some comments on the text:
L126: It is not necessary to indicate under each table, this „Quoirin and Lepoivre [23], modified by Aitken-142 Christie et al. [24]“ or this „supplemented with 6-benzyladenine (BA at 4.4, 22 and 44 μM), especially since the graph shows the concentrations.
L405: Material and Methods to be revised, clearly distinguishing the separate processes of in vitro propagation: Sterilization, Organogenetic processes, Rooting and Adaptation. Clearly and precisely describe the materials used, without repetition. More accurate differentiation of MM and Results
L406: Please, indicate the age of the plants
L444: To avoid repetitions of the type of: “The pH of the medium was adjusted 444 to 5.8 before autoclaving (121 °C, 20 min)”, it is sufficient to specify it once, but in an appropriate place.
L531: The Statistical Analysis should be shaped in a similar way, to avoid repetitions such as these “Contamination, survival and explants forming shoots (EFS) percentages for each condition tested were measured after two months of culture”, аlthough the experiments are separate
Author Response
Dear Reviewer,
Enclosed you can find the response to your comments and suggestions.
Thank you very much.
Best regards.

Reviewer 2 Report
The publication covers all steps of micropropagation of very difficult and essential Pinus ponderosa. The manuscript is written correctly according to the standards of scientific publications.
I marked just some unclear words to correct in the Abstract (which should be independently understood part).
I have just one scientific question considering the plan of the experiment.
Why authors did not decide to test also mixed Red and Blue LED light?
This mixture is a standard now in scientific as well as commercial in vitro labs and in the whole greenhouse industry. Such a mixture gives the results the most comparable to fluorescent light (if applied at similar intensity).

Author Response

(The authors gave the same response as above.)

Reviewer 3 Report
The manuscript by Rojas-Vargas et al., on Pinus ponderosa shows the development of a method for shoot regeneration from zygotic embryos generating a factor 3-4 using BA. Light quality appears to affect stages of in vitro growth. The results are clearly described, the discussion shows a comparison with existing literature. Overall the work is straightforward, and some improvement of the methods description will be needed, as I clarify below.
1) in view of the difference between White LEDs and fluorescent light in table 1 for rooting, it is necessary to introduce the composition of the light (in wavelength bands) for these two light sources, as they both qualify as white. Then, a discussion why the difference may occur is also warranted.
2) line 199: the rationale for the work on morphological characterisation can be expressed at the beginning of paragraph 2.1.
3) line 230: the irregular pattern of stomata should be shown somewhere, or clarified.
4) line 406: seeds from a clonal seed orchard: what does this mean? what is the genetic diversity of the seeds used? Also, how were they harvested? (on the tree? various trees? etc...) Should we expect that the results are typical for this particular clone, or are they representative for the entire species?
Author Response

(The authors gave the same response as above.)
